# Coherent DOA Estimation Algorithm with Co-Prime Arrays for Low SNR Signals

**DOI:** 10.3390/s23239320

**Published:** 2023-11-22

**Authors:** Fan Zhang, Hui Cao, Kehao Wang

**Affiliations:** School of Information Engineering, Wuhan University of Technology, Wuhan 430070, China; zhangfan99@whut.edu.cn (F.Z.); kehao.wang@whut.edu.cn (K.W.)

**Keywords:** DOA estimation, space-time correlation matrix, spatial smoothing, low SNR

## Abstract

The Direction of Arrival (DOA) estimation of coherent signals in co-prime arrays has become a popular research topic. However, traditional spatial smoothing and subspace algorithms fail to perform well under low signal-to-noise ratio (SNR) and small snapshots. To address this issue, we have introduced an Enhanced Spatial Smoothing (ESS) algorithm that utilizes a space-time correlation matrix for de-noising and decoherence. Finally, an Estimating Signal Parameter via Rotational Invariance Techniques (ESPRIT) algorithm is used for DOA estimation. In comparison to other decoherence methods, when the SNR is −8 dB and the number of snapshots is 150, the mean square error (MSE) of the proposed algorithm approaches the Cramér–Rao bound (CRB), the probability of resolution (PoR) can reach over 88%, and, when the angular resolution is greater than 4°, the estimation accuracy can reach over 90%.

## 1. Introduction

Array signal processing is a specialized domain within the broader field of signal processing. It primarily focuses on utilizing antenna arrays to mathematically model received signals, subsequently forming matrices for further analysis. One significant branch and key research area in array signal processing is Direction of Arrival (DOA) estimation, which has a wide range of applications, including automotive systems, nondestructive testing, and radar [1,2,3,4,5], among others. Depending on the arrangement of antenna arrays, various array models can be classified into categories such as Uniform Linear Array (ULA) [6,7], Uniform Circular Array (UCA) [8,9], and Non-Uniform Linear Array (NULA) [10,11].

The Multiple Signal Classification (MUSIC) algorithm [12] is a prominent traditional method for DOA estimation. By leveraging the orthogonality between signal and noise subspaces, it constructs a spatial spectral function and estimates signal parameters through spectral peak searches. This milestone signifies a thriving era for spatial spectrum estimation direction finding. Although the algorithm provides accurate estimations, it is complex and time-consuming. To mitigate complexity, Barabell introduced the ROOT-MUSIC algorithm [13], which employs Pisarenko decomposition as its fundamental concept. This approach enables precise signal angle estimation without the need for spectral peak searches. Furthermore, the Estimating Signal Parameter via Rotational Invariance Techniques (ESPRIT) [14] algorithm, based on rotation invariance technology, utilizes the rotation invariance of signal subspace for signal parameter estimation, offering lower complexity compared to MUSIC. As classic signal subspace algorithms, both ESPRIT and MUSIC have inspired generations of researchers.

In recent years, the co-prime array [15] has gained significant attention due to its high degree of freedom. In the field of covariance arrays, Huang [16] and his team assumed the presence of non-uniform noise in the received signals. They reconstructed the covariance matrix using virtual array interpolation, matrix completion, and linear prediction, followed by the application of traditional DOA estimation methods to the reconstructed matrix for angle estimation. In situations involving non-redundant sensor faults, Sun [17] and his colleagues suggested employing the Singular Value Threshold (SVT) algorithm for matrix completion to fill the gaps, which has proven to be effective in DOA estimation. Shi [18] and his colleagues proposed an inverse discrete Fourier transform (IDFT)-based DOA estimation algorithm for co-prime array, where both DOAs and the power of the sources can be efficiently estimated with an increased number of degrees-of-freedom. Additionally, they presented a fast Fourier transform (FFT)-based DOA estimation algorithm [19] for a co-prime multiple-input multiple-output (MIMO) radar, where both the performance and the system overhead are well balanced. Furthermore, they introduced the framework of co-prime array signal processing into massive MIMO systems and proposed an efficient IDFT-based DOA estimation algorithm [20].

In any array model, it is impossible to achieve a completely ideal signal. To better implement DOA algorithms in real-life scenarios, an increasing number of researchers are focusing on studying coherent signals. When a coherent signal is present in the received signal, decoherence becomes essential for accurately estimating DOA. Currently, the primary decoherence methods can be categorized into two types: dimensionality reduction and non-dimensionality reduction. Dimensionality reduction mainly consists of spatial smoothing and matrix reconstruction algorithms, such as the forward-only spatial smoothing (FOSS) algorithm [21] and the matrix decomposition algorithm. On the other hand, non-dimensionality reduction primarily involves frequency smoothing [22] algorithms and the Toeplitz method [23]. Wu et al. [24] suggested using Forward/Backward Spatial Smoothing (FBSS) [25] and the MUSIC algorithm to process coherent signals in the co-prime array. By preprocessing the received signals with FBSS and searching for overlapping peaks of the MUSIC spectrum in the two sub-arrays, DOA can be accurately estimated. However, this algorithm has high complexity and requires a significant amount of time for estimation. Tang et al. [26] proposed using the principal singular vectors of multiple-Toeplitz matrices to enhance the accuracy of DOA estimation. By employing the Toeplitz matrix, an equivalent covariance matrix was obtained, and a linear equation was constructed using the relationship between the Toeplitz matrix and the signal subspace. Accurate DOA estimation could be achieved by solving this linear equation.

This paper focuses on the study of coherent signals in co-prime arrays, specifically on how to accurately estimate DOA accurately in the case of low signal-to-noise ratio (SNR) and small snapshots. The practical application of the model lies in the field of medical imaging, where it enhances the resolution and quality of various imaging techniques, including ultrasonic imaging and nuclear magnetic resonance imaging [16]. Drawing inspiration from the literature [27,28], we proposed an Enhanced Spatial Smoothing (ESS) algorithm based on the space-time correlation matrix. First, it is necessary to de-noise the received signal. Based on the theoretical foundation of array signal processing, appropriate delay processing is applied to the received signal to obtain the space-time correlation matrix, which serves the purpose of de-noising. Next, the de-noised signals need to undergo a decoherence process. The ESS algorithm is then used to decohere part of the space-time correlation matrix, obtaining the covariance matrix of the two sub-arrays that compose the co-prime array. Finally, the covariance matrix of the two sub-arrays is estimated using the ESPRIT to estimate the DOA. Simulation results demonstrate that, compared to the FBSS and ESS algorithms, when the SNR is −8 dB and the number of snapshots is 150, the mean square error (MSE) of the proposed algorithm approaches Cramér–Rao bound (CRB), the probability of resolution (PoR) can reach over 88%, and, when the angular resolution is greater than 4°, the estimation accuracy can reach over 90%.

Throughout the paper, we use the lowercase (uppercase) boldface symbols to represent vectors (matrices). ${\left(\cdot \right)}^{T}$ and ${\left(\cdot \right)}^{H}$ denote the transpose and the conjugate transpose, respectively. $I_{N}$ denotes N×N identity matrix, $\mathrm{diag}\left(\cdot \right)$ denotes the diagonal matrix operator, and $E\left(\cdot \right)$ denotes the expectation operator, ${\left(\cdot \right)}^{†}$ denotes the Moore–Penrose Inverse.

## 2. Signal Model

The co-prime array is a recently proposed array structure designed to enhance the array’s degrees of freedom. Utilizing the characteristics of co-prime numbers, it is a sparse array composed of two ULAs with sub-arrays element counts M and N, where M and N are co-prime integers. The spacing between the two sub-arrays is Nλ∕2 and Mλ∕2, respectively, with λ representing the wavelength of the received signal. Figure 1 illustrates the arrangement of the co-prime array and its two sub-arrays.

Figure 1a illustrates the general structure of the co-prime array. As the two sub-arrays share the first element as a reference point, the total number of elements is M+N−1. To facilitate analysis, the co-prime array can be divided into two ULAs, as depicted in Figure 1b. As observed from Figure 1, the spacing between antennas can be categorized into:(1)$$d=\left\{mN\left(\frac{\lambda }{2}\right)\cup \;nM\left(\frac{\lambda }{2}\right)\right\}$$
where 0≤m≤M−1,0≤n≤N−1.

Assuming that there are far-field narrow-band signals arriving at the co-prime array from various directions in space, the incident angles of these signals can be denoted as $\theta_{k}$, where k=1,2,3,⋯,K. The noise present is additive white Gaussian noise (AWGN). Under these conditions, the received signal can be modeled as follows:(2)$$x\left(t\right)=A\left(\theta \right)s\left(t\right)+n\left(t\right)$$
where $A\left(\theta \right)={\left[A_{N}\left(\theta \right),A_{M}\left(\theta \right)\right]}^{T}$ represents the $\left(M+N-1\right)\times K$ array steering matrix with $a\left(\theta_{k}\right)={\left[a_{N}\left(\theta_{k}\right),a_{M}\left(\theta_{k}\right)\right]}^{T}$, $a_{N}\left(\theta_{k}\right)={\left[1,e^{-jM\pi \sin\left(\theta_{k}\right)},\ldots ,e^{-jM\left(N-1\right)\pi \sin\left(\theta_{k}\right)}\right]}^{T}$, $a_{M}\left(\theta_{k}\right)={\left[1,e^{-jN\pi \sin\left(\theta_{k}\right)},\ldots ,e^{-jN\left(M-1\right)\pi \sin\left(\theta_{k}\right)}\right]}^{T}$, respectively, $s\left(t\right)={\left[s_{1}\left(t\right),s_{2}\left(t\right),\ldots ,s_{K}\left(t\right)\right]}^{T}$ is a K×1 signal vector, and $n\left(t\right)={\left[n_{1}\left(t\right),n_{2}\left(t\right),\ldots ,n_{\left(M+N\right)}\left(t\right)\right]}^{T}$ denotes the $\left(M+N-1\right)\times 1$ complex Gaussian noise vector, which is assumed to be uncorrelated to signals with a zero mean and variance $\delta^{2}$.

According to the definition of covariance matrix [24], the array output covariance matrix can be expressed as:(3)$$R_{xx}\left(t\right)=E\left[x\left(t\right)x^{H}\left(t\right)\right]=AR_{s}A^{H}+\sigma^{2}I_{M+N}$$
where $R_{s}=E\left[s\left(t\right)s^{H}\left(t\right)\right]$ is the signal covariance matrix.

## 3. Proposed Approach

In this study, we consider the scenario where the correlated signal from the $\theta_{k}$ direction $\left(k=1,2,3,\cdots ,K\right)$ impacts the co-prime array, and the correlation coefficient is represented as $\alpha_{K\times 1}$. When $s\left(t\right)$ is a constant, Equations (2) and (3) can be reformulated as:(4)$$x\left(t\right)=A\left(\theta \right)\alpha s\left(t\right)+n\left(t\right)=\left(\begin{array}{c} x_{N}\left(t\right)\\ x_{M}\left(t\right) \end{array}\right)+n\left(t\right)$$
(5)$$R_{xx}\left(t\right)=E\left[x\left(t\right)x^{H}\left(t\right)\right]=R_{xs}\left(t\right)+R_{xn}\left(t\right)=\left[\begin{array}{cc} x_{N}x_{N}^{H} & x_{N}x_{M}^{H}\\ x_{M}x_{N}^{H} & x_{M}x_{M}^{H} \end{array}\right]+\sigma^{2}I_{M+N}\;=\left[\begin{array}{cc} R_{NN} & R_{NM}\\ R_{MN} & R_{MM} \end{array}\right]+\sigma^{2}I_{M+N}$$

Subsequently, it is necessary to de-noise and de-correlate the coherent signals. The conventional FBSS algorithm exhibits effective de-correlation performance, however, it struggles to suppress noise under low SNR conditions. To enhance the de-correlation capabilities of the algorithm and address the issue of noise reduction in low SNR situations, this study proposes a space-time correlation algorithm based on improved spatial smoothing.
*A.* *Space-Time correlation algorithm*

The signal exhibits strong correlation in both the time and space domains, whereas the noise demonstrates weak correlation in these domains. This characteristic of the signal can be exploited for de-noising purposes. By reconstructing the acceptance matrix of the co-prime array using the space-time correlation matrix, it is possible to reduce the noise energy relatively, thereby enhancing the SNR and achieving the objective of noise reduction.

Initially, for any given delay τ>0, the space-time correlation matrix for the entire co-prime array can be represented as [24]:(6)$${\tilde{R}}_{xx}\left(\tau \right)=E\left[x\left(t-\tau \right)x^{H}\left(t+\tau \right)\right]$$

Upon simplification, the space-time correlation matrix can be reduced to the subsequent form [24]:(7)$${\tilde{R}}_{xx}\left(\tau \right)=A\alpha E\left[s\left(t-\tau \right)s^{H}\left(t+\tau \right)\right]\alpha^{H}A^{H}+E\left[n\left(t-\tau \right)n^{H}\left(t+\tau \right)\right]$$

As $n\left(t\right)={\left[n_{1}\left(t\right),n_{2}\left(t\right),\ldots n_{\left(M+N\right)}\left(t\right)\right]}^{T}$ represents a complex Gaussian white noise vector, the noise component can be considered insignificant. Based on array signal processing theory, the narrowband far-field signal satisfies the following conditions [27]:(8)$$s\left(t+\tau \right)\approx s\left(t\right)e^{j\omega \tau }$$
where ω represents the carrier frequency.

By substituting Equation (8) into Equation (7), the space-time correlation matrix can be formulated as follows [29]:(9)$${\tilde{R}}_{xx}\left(\tau \right)\approx A\alpha R_{s}\alpha^{H}A^{H}e^{-j2\omega \tau }=R_{xs}\left(t\right)e^{-j2\omega \tau }$$
where $R_{xs}\left(t\right)=A\alpha R_{s}\alpha^{H}A^{H}$.

Second, to eliminate $e^{j\omega \tau }$, we employ the covariance technique to obtain the covariance matrix without the $e^{j\omega \tau }$ term [27]:(10)$${\tilde{R}}_{ST}\left(\tau \right)={\tilde{R}}_{xx}\left(\tau \right){\tilde{R}}^{H}{}_{xx}\left(t\right)\approx R_{xs}\left(t\right)R_{xs}^{H}\left(t\right)=\left[\begin{array}{cc} R_{NN} & R_{NM}\\ R_{MN} & R_{MM} \end{array}\right]{\left[\begin{array}{cc} R_{NN} & R_{NM}\\ R_{MN} & R_{MM} \end{array}\right]}^{H}\;=\left[\begin{array}{cc} R_{NN}R_{NN}^{H}+R_{NM}R_{NM}^{H} & R_{NN}R_{MN}^{H}+R_{NM}R_{MM}^{H}\\ R_{MN}R_{NN}^{H}+R_{MM}R_{NM}^{H} & R_{MN}R_{MN}^{H}+R_{MM}R_{MM}^{H} \end{array}\right]$$

After the noise removal, we apply the ESS algorithm to achieve decoherence.
*B.* *Enhanced Spatial Smoothing*

The conventional smoothing algorithm is unable to fully utilize the sub-array information, which leads us to propose the ESS algorithm. Similar to the FBSS technique, the ESS algorithm addresses coherence issues by employing smoothing methods, but with distinct covariance matrices. As illustrated in Figure 2, a ULA consisting of D elements can be divided into C sub-arrays, each containing O elements. The parameters D, O, and C are interconnected through the following equation:(11)C=D−O+1

The covariance matrix of the ESS algorithm for rank recovery data is given by [28]:(12)$$R_{ESS}=\frac{1}{2C}\sum_{i=1}^{C}\sum_{j=1}^{C}\left[\left(R_{ii}R_{jj}+{\bar{R}}_{ii}{\bar{R}}_{jj}\right)+\left(R_{ij}R_{ji}+{\bar{R}}_{ij}{\bar{R}}_{ji}\right)\right]$$
(13)$$R_{ij}=J_{i}RJ_{j}$$
(14)$${\bar{R}}_{ij}=JR_{ij}J$$

In the ESS processing, the matrix R is utilized, where $J_{i}=\left[0_{O\times \left(i-1\right)}\;I_{O}\;0_{O\times \left(M+N-i-O+1\right)}\right]$, $J_{j}=\left[0_{\left(j-1\right)\times O}\;I_{O}\;0_{\left(M+N-j-O+1\right)\times O}\right]$, and J represents the $\left(L\times L\right)$ anti-identity matrix.

As illustrated in Appendix A, matrices $\left(R_{NN}R_{NN}^{H}+R_{NM}R_{NM}^{H}\right)$ and $\left(R_{MN}R_{MN}^{H}+R_{MM}R_{MM}^{H}\right)$ in ${\tilde{R}}_{ST}$ can be considered as the output covariance matrices of a ULA with array numbers N and M, respectively. We applied the ESS algorithm to remove coherence between them, resulting in two sub-arrays with sub-array elements N and M. The covariance matrices of these sub-arrays are denoted as $R_{M}$ and $R_{N}$, respectively. Finally, we conducted DOA estimation on these sub-arrays.
*C.* *DOA estimation by subspace algorithm*

In this study, we employ ST and ESS to achieve noise reduction and decoherence, respectively. Given the similar algorithm principles of the two sub-arrays, we only introduce the subspace algorithm of $R_{M}$ for DOA estimation.

In Equation (11), assuming O=M, we can obtain the decohered $R_{M}$ and perform eigenvalue decomposition on it [25]:(15)$$R_{M}=U_{s}\Lambda_{s}U_{s}^{H}+U_{n}\Lambda_{n}U_{n}^{H}$$

In this case, the corresponding eigenvalues of the *K* largest eigenvectors of $R_{M}$ form the signal subspace $U_{S}$. As $U_{S}$ and $A_{M}$ span the same subspace, a non-singular matrix T exists, such that:(16)$$A_{M}\left(\theta \right)=U_{s}T$$

The shift invariance in ESPRIT can be rephrased as follows:(17)$${\underline{A}}_{M1}\left(\theta \right)={\bar{A}}_{M1}\left(\theta \right)\Phi$$
(18)$$\left\{\begin{array}{c} {\underline{A}}_{M1}\left(\theta \right)={\underline{J}}_{1}A_{M}\left(\theta \right)\\ {\bar{A}}_{M1}\left(\theta \right)={\bar{J}}_{1}A_{M}\left(\theta \right) \end{array}\right.$$
where ${\underline{J}}_{1}=\left[0_{\left(M-1\right)\times 1}\;I_{M-1}\right]$, ${\bar{J}}_{1}=\left[I_{M-1}\;0_{\left(M-1\right)\times 1}\right]$ and $\Phi =diag\left(e^{-jN\pi \sin\left(\theta_{1}\right)},\cdots ,e^{-jN\pi \sin\left(\theta_{k}\right)}\right)$ are given, the equation for rotational invariance (RIE) can be expressed as:(19)$${\underline{U}}_{s1}\left(\theta \right)={\bar{U}}_{s1}\left(\theta \right)\Psi$$
(20)$$\left\{\begin{array}{c} {\underline{U}}_{s1}\left(\theta \right)={\underline{J}}_{1}U_{s}\\ {\bar{U}}_{s1}\left(\theta \right)={\bar{J}}_{1}U_{s} \end{array}\right.$$

Using Equation (19) and given values of $\Psi =T\Phi T^{-1}$, we can compute Ψ. With the provided value of Φ, we can then obtain the estimated value $\hat{T}$ of T. By substituting this value into Equation (16), we can determine the estimated value of $A_{M}$:(21)$${\hat{A}}_{M}\left(\theta \right)=U_{s}\hat{T}$$

Despite noise reduction, some noise remains in the signal, causing a discrepancy between the estimated value of $A_{M}$ in Equation (21) and the true value. The magnitude of this error is:(22)$${\hat{A}}_{M}\left(\theta \right)=A_{M}\left(\theta \right)\Gamma \Sigma +\Delta A_{M}$$
given Γ as a permutation matrix and Σ as a diagonal scaling matrix with $\Sigma =diag\left(\gamma_{M1},\gamma_{M2},\cdots ,\gamma_{MK}\right)$, let the element of matrix ${\hat{A}}_{M}\left(\theta \right)$ be represented by ${\hat{a}}_{Mk}$. Then [29]:(23)$$\underset{\theta_{Mk},\gamma_{Mk}}{\mathrm{Min}}{\left\|{\hat{a}}_{Mk}-\gamma_{Mk}a_{Mk}\right\|}_{2}^{2},\forall k=1,2,\ldots ,K.$$

We can approximate the value of ${\hat{\gamma }}_{Mk}$ and then apply Newton’s method to derive the ML formulation for $\theta_{Mk}$ [29]:(24)$${\hat{\gamma }}_{Mk}={\left(a_{Mk}\right)}^{†}{\hat{a}}_{Mk}$$
(25)$${\hat{\theta }}_{Mk}=arg\;\underset{\theta_{Mk}}{\max}{\left|a_{Mk}^{H}{\hat{a}}_{Mk}\right|}^{2},\forall k=1,2,\ldots ,K.$$
as $a_{Mk}^{H}{\hat{a}}_{Mk}=M$ is constant, it is neglected. Hence, $\theta_{Mk}$ is updated by
(26)$$\theta_{Mk}^{\left(l+1\right)}=\theta_{Mk}^{\left(l\right)}-\mu^{\left(l\right)}h^{-1}\left(\theta_{Mk}^{\left(l\right)}\right)g\left(\theta_{Mk}^{\left(l\right)}\right)$$
(27)$$h\left(\theta_{Mk}\right)=2ℜ\left({\ddot{a}}_{Mk}^{H}{\hat{a}}_{Mk}{\hat{a}}_{Mk}^{H}a_{Mk}+{\overset{˙}{a}}_{Mk}^{H}{\hat{a}}_{Mk}{\hat{a}}_{Mk}^{H}{\overset{˙}{a}}_{Mk}\right)$$
(28)$$g\left(\theta_{Mk}\right)=2ℜ\left({\overset{˙}{a}}_{Mk}^{H}{\hat{a}}_{Mk}a_{Mk}^{H}{\hat{a}}_{Mk}\right)$$
in this case, ℜ represents the real part, while ${\overset{˙}{a}}_{Mk}=\frac{\partial a\left(\theta_{Mk}\right)}{\partial \theta_{Mk}}$ and ${\ddot{a}}_{k}=\frac{\partial^{2}a\left(\theta_{Mk}\right)}{\partial \theta_{Mk}^{2}}$ denote the other components.

After acquiring the angle estimation for the two sub-arrays, we can obtain the estimated angle of the co-prime array by calculating their average value:(29)$${\hat{\theta }}_{k}=\frac{{\hat{\theta }}_{Mk}+{\hat{\theta }}_{Nk}}{2}$$

To summarize, the suggested Algorithm 1 can be outlined as follows:
**Algorithm 1**. DOA Estimation of Coherent Signal in Co-prime Array.**Step 1**  Process the space-time correlation for the received signal and obtain the space-time correlation matrix ${\tilde{R}}_{ST}\left(\tau \right)$ using Equations (6) and (10).**Step 2**  Apply ESS to the space-time correlation matrix ${\tilde{R}}_{ST}\left(\tau \right)$ twice using Equation (12) to obtain two covariance sub-matrices, $R_{M}$ and $R_{N}$.**Step 3**  Use the traditional ESPRIT algorithm to process $R_{M}$ and $R_{N}$ separately to obtain the estimated value of the array guidance matrix ${\hat{A}}_{M}\left(\theta \right)$ according to Equation (21).**Step 4**  Obtain the angle estimates of the two sub-arrays in the co-prime array, ${\hat{\theta }}_{Mk}$ and ${\hat{\theta }}_{Nk}$, using Equation (25).
**Step 5**  Use Equation (29) to obtain the angle estimate ${\hat{\theta }}_{k}$ of the coherent signal in the co-prime array.

## 4. Simulation Results

In this chapter, the primary focus is to evaluate the performance of the proposed algorithm through numerical simulations and compare it with existing algorithms.

Assuming that the number of receiving antennas in the co-prime array is M=7 and N=5, two coherent signals with equal power are considered to be white Gaussian noise and incident on the array from $\theta_{1}=3{}^{\circ}$ and $\theta_{2}=8{}^{\circ}$, with a correlation coefficient of $\alpha ={\left[1,e^{j\pi /6}\right]}^{T}$. The algorithm’s performance is analyzed using SNR=−2 dB, delay τ=1, the number of snapshots L=600, and the number of Monte Carlo tests I=200. The MSE [14] and PoR [28] are defined as follows:(30)$$MSE=\frac{1}{IK}\sum_{i=1}^{I}\sum_{k=1}^{K}{\left({\hat{\theta }}_{k}\left(i\right)-\theta_{k}\right)}^{2}$$
(31)$$PoR=\left(\frac{1}{IK}\sum_{k=1}^{K}N_{k}\right)\times 100\%$$
(32)$$N_{k}=\left\{\begin{array}{c} 0\;\left|{\hat{\theta }}_{k}-\theta_{k}\right|\geq 0.1\\ 1\;\left|{\hat{\theta }}_{k}-\theta_{k}\right|<0.1 \end{array}\right.$$
where ${\hat{\theta }}_{k}\left(i\right)$ represents the estimate of $\theta_{k}$ for the ith trial, and $N_{k}$ is used to count the number of successful resolutions of the *k*th signal.

Among numerous bounds for DOA estimation, the local CRB is only tight asymptotically. To better evaluate hybrid coherent/incoherent multi-source DOA estimate, we introduce explicit Ziv-Zakai bound (ZZB) [30] for evaluation. The specific performance comparison is shown in Figure 3.

Figure 3 shows that ZZB is 3 dB away from CRB. This may be due to in the ZZB algorithm the precondition of effective angular separation is greater than 10° [30], while in this article, we set an angular separation of less than 5°. Therefore, in the subsequent performance comparison, we will use CRB for performance evaluation.

Afterward, we compare the proposed algorithm with the FBSS algorithm and the ESS algorithm.
*A.* *MSE and PoR versus SNR*

When only the *SNR* is changed, varying from −14 dB to 8 dB, the simulation results can be observed in Figure 4 and Figure 5.

As depicted in Figure 4 and Figure 5, it is evident that the accuracy of the three algorithms improves with the increase of SNR. The MSE of the three algorithms decreases, and the performance of the proposed algorithm is superior to that of the FBSS and ESS algorithms under low SNR conditions (from −14 dB to −2 dB). Under high SNR conditions (from −2 dB to 8 dB), the proposed algorithm outperforms FBSS and is slightly better than the ESS algorithm. The proposed algorithm can accurately estimate DOA at -8 dB, while ESS and FBSS can accurately estimate at −6 dB and −4 dB, respectively. Thus, compared to FBSS and ESS, the proposed algorithm can accurately estimate at a lower SNR. When the three algorithms reach the inflection point, they all approach CRB. When SNR=0 dB, the accuracy of the three algorithms can reach 100%. Simultaneously, under the same SNR condition, the accuracy of the proposed algorithm is higher than that of FBSS and ESS. When SNR=−6 dB, the accuracy of the proposed algorithm can reach 95%.

In conclusion, compared to FBSS and ESS, the proposed algorithm can reach the inflection point for accurate estimation more quickly and exhibits better noise reduction and solution correlation under low SNR conditions.
*B.* *MSE and PoR versus snapshots*

When only the parameter L is altered, with its range varying from 100 to 700, the simulation results can be observed in Figure 6 and Figure 7.

As illustrated in Figure 6 and Figure 7, it is evident that as the number of snapshots L increases, the MSE of the three algorithms gradually decreases, eventually becoming parallel to the CRB, and the estimation accuracy correspondingly increases. The inflection points for FBSS, ESS, and the proposed algorithm are 250, 200, and 150, respectively. Furthermore, the MSE of the proposed algorithm remains lower than that of FBSS and ESS after reaching the inflection point. Additionally, the accuracy of the proposed algorithm surpasses that of FBSS and ESS, regardless of the number of snapshots L. In conclusion, under low SNR conditions, the proposed algorithm can accurately estimate with a lower number of snapshots.
*C.* *MSE and PoR versus angular separation*

When only the angular separation is altered, with the two coherent signals received from $\left[3{}^{\circ},3{}^{\circ}+\Delta \theta \right]$, where Δθ ranges from 3.5° to 5°, the simulation results can be observed in Figure 8 and Figure 9.

Figure 8 and Figure 9 demonstrate that as the angular separation gradually increases, the MSE of the three algorithms progressively decreases, while the PoR correspondingly increases until it reaches 1. Throughout this process, the MSE of the proposed algorithm consistently remains lower than that of FBSS and ESS, and the PoR is consistently higher than that of FBSS and ESS. Notably, when the angular separation lies between 3.5° and 4°, the MSE curve of the proposed algorithm exhibits a steeper decline compared to those of FBSS and ESS, indicating that the proposed algorithm possesses superior estimation accuracy and angular resolution. Particularly, when the angular separation exceeds 4.6°, the estimation error of the proposed method achieves a relatively stable state, and when the angular resolution is greater than 4°, the estimation accuracy can reach over 90%.
*D.* *Algorithm complexity analysis*

In this section, the complexity analysis of the algorithm is divided into two aspects. The first aspect compares the relationship between the number of antennas and the algorithm’s execution time, while the second aspect focuses on numerical complexity analysis. The simulation results concerning the number of antennas and the algorithm’s running time can be observed in Figure 10 and Table 1. The algorithmic complexity of the three algorithms is presented in Table 2.

As observed in Figure 10 and Table 1 and Table 2, the execution time of the proposed algorithm is slightly longer than that of FBSS and ESS. However, even when M=97, the difference between the proposed algorithm and the FBSS, which has the shortest running time, is merely 0.78 × 10^−3^ s. The execution time of the proposed algorithm is 150% that of FBSS and ESS. Considering the performance improvement offered by the proposed algorithm, it is reasonable to accept a minor increase in execution time.

## 5. Conclusions

In this study, we address the challenge of accurately estimating DOA when the received signal of a co-prime array is coherent. We propose an ESS algorithm based on the space-time correlation matrix. The space-time correlation matrix is initially employed for noise reduction, followed by the application of the ESS algorithm for decoherence. Lastly, the ESPRIT algorithm is utilized for DOA estimation. Simulation results indicate that the proposed algorithm can effectively estimate DOA under low SNR conditions and small snapshots. When SNR=−8 dB and the number of snapshots L=600, the MSE of the proposed algorithm approaches CRB, the PoR can reach over 88%, and when the angular resolution is greater than 4°, the estimation accuracy can reach over 90%. Nevertheless, the algorithm has certain drawbacks. Although its complexity and execution time are comparable to traditional algorithms, they are still higher. Future improvements can be made to address these limitations.

## Figures and Tables

**Figure 1 sensors-23-09320-f001:**
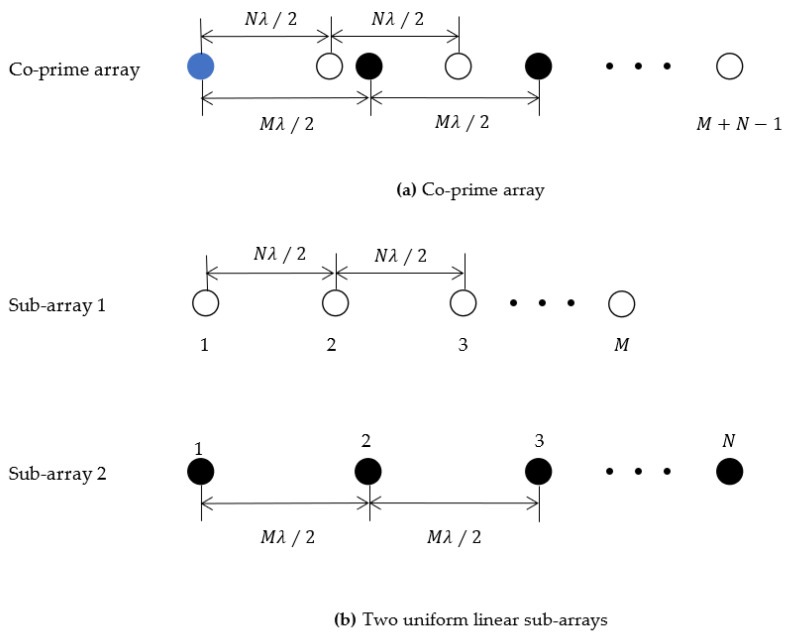
Basic structure of co-prime linear array.

**Figure 2 sensors-23-09320-f002:**
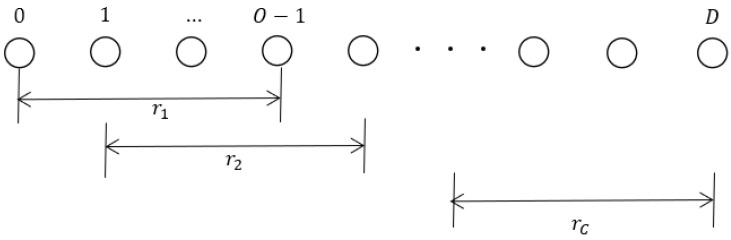
Overlapping sub-arrays using the smoothing technique for co-prime arrays.

**Figure 3 sensors-23-09320-f003:**
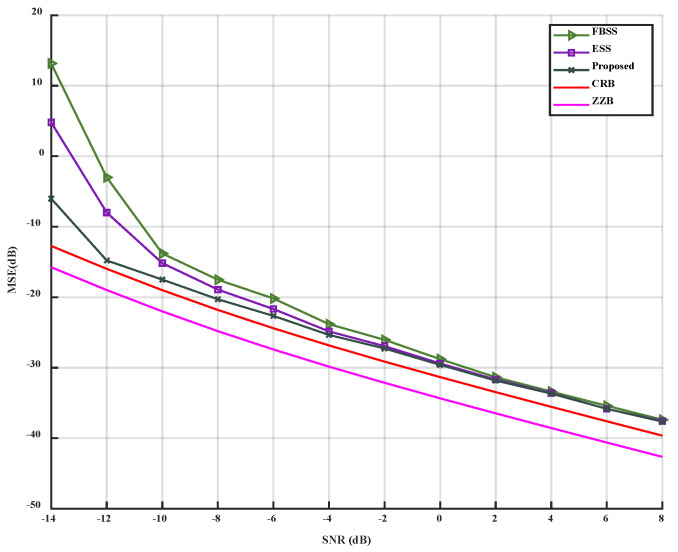
Performance comparison between MSE and ZZB.

**Figure 4 sensors-23-09320-f004:**
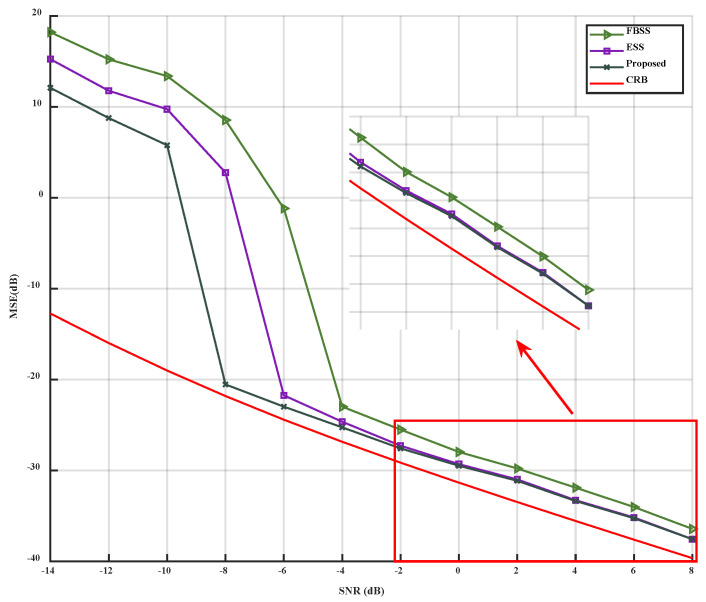
MSE versus SNR.

**Figure 5 sensors-23-09320-f005:**
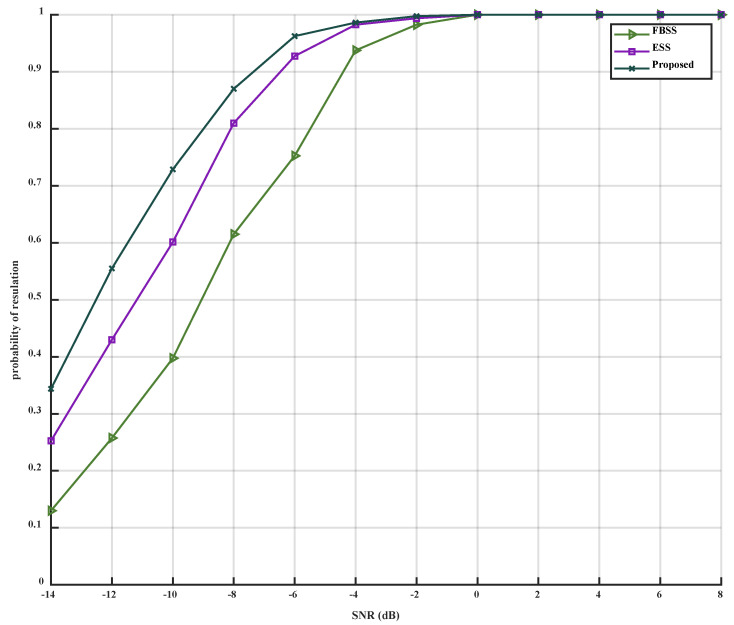
PoR versus SNR.

**Figure 6 sensors-23-09320-f006:**
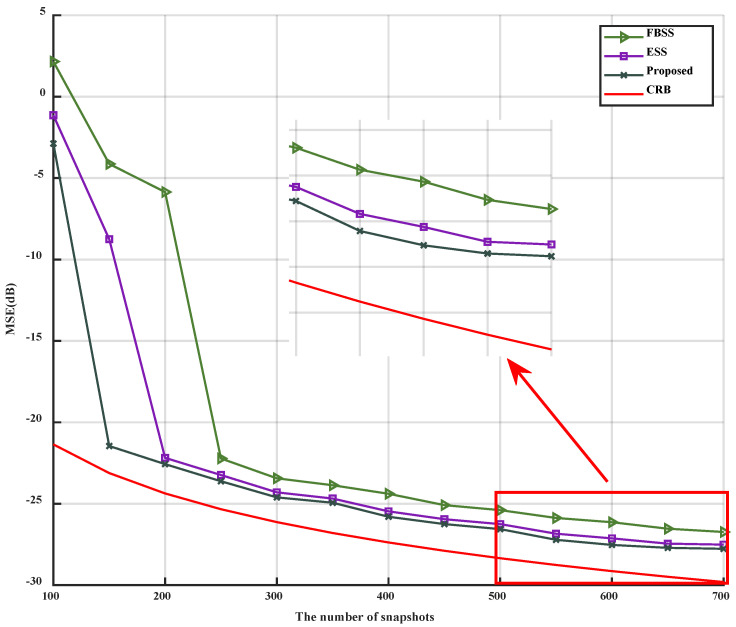
MSE versus snapshots.

**Figure 7 sensors-23-09320-f007:**
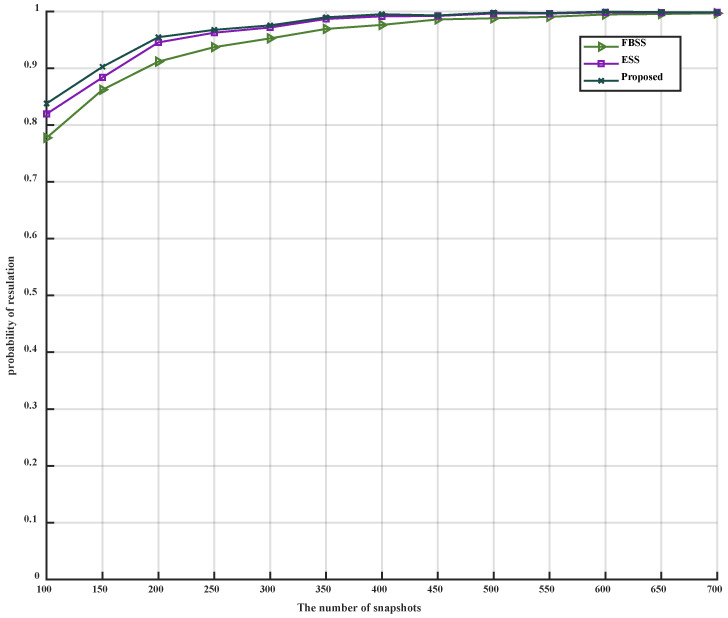
PoR versus snapshots.

**Figure 8 sensors-23-09320-f008:**
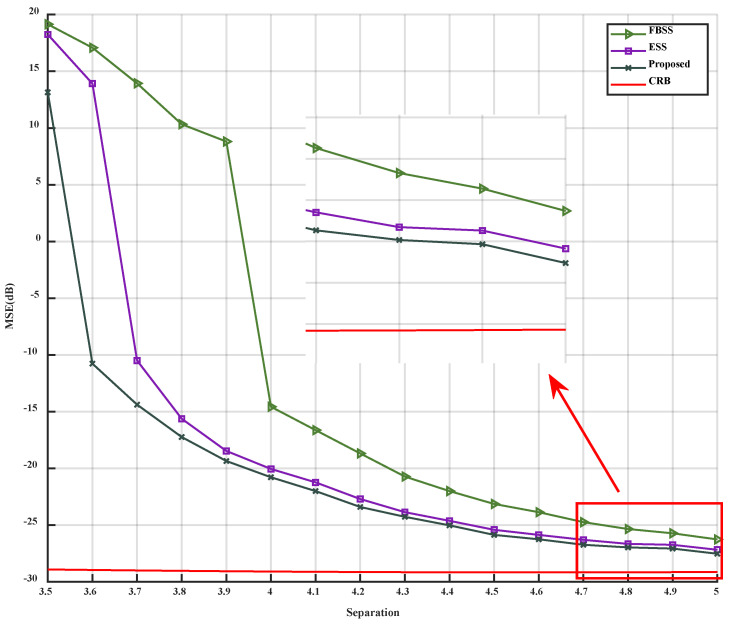
MSE versus angular separation.

**Figure 9 sensors-23-09320-f009:**
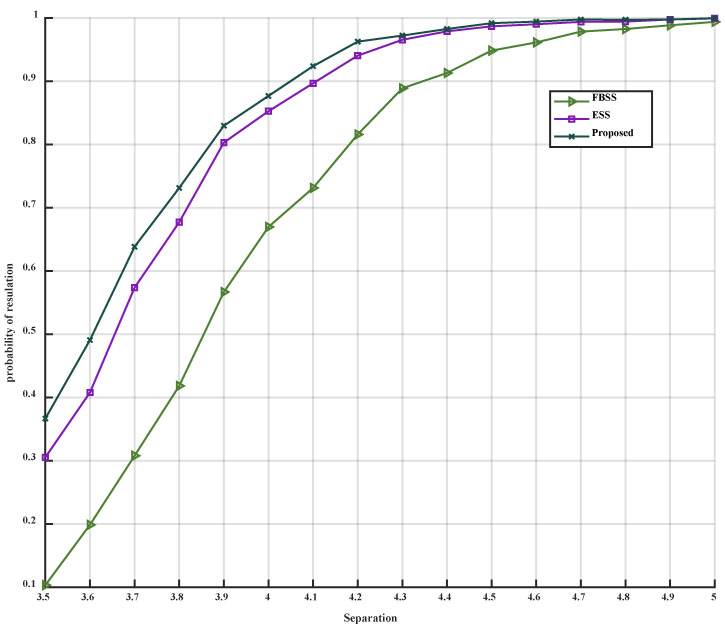
PoR versus angular separation.

**Figure 10 sensors-23-09320-f010:**
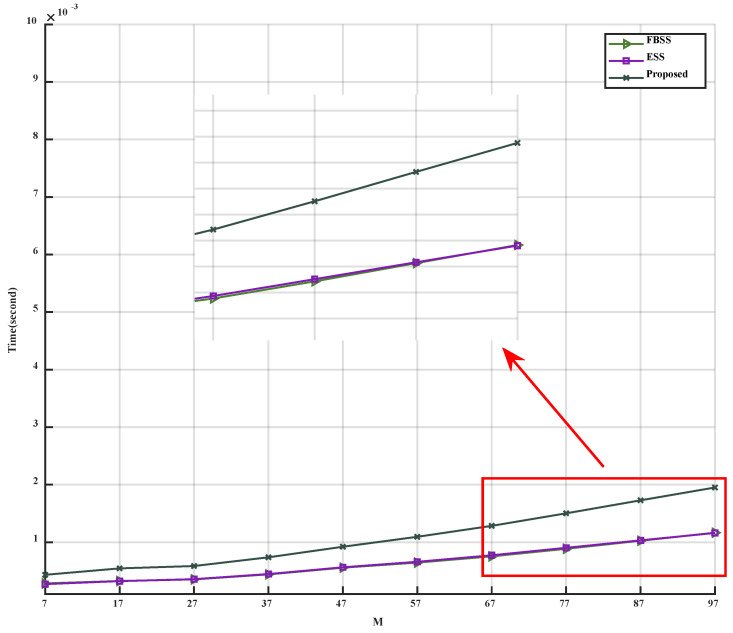
Time versus number of antennas.

**Table 1 sensors-23-09320-t001:** Time versus number of antennas.

		M	7	17	⋯	87	97
	Time (s)						
Algorithm							
FBSS			0.28 × 10^−3^	0.33 × 10^−3^	⋯	1.02 × 10^−3^	1.17 × 10^−3^
ESS			0.27 × 10^−3^	0.32 × 10^−3^	⋯	1.03 × 10^−3^	1.16 × 10^−3^
Proposed			0.43 × 10^−3^	0.55 × 10^−3^	⋯	1.73 × 10^−3^	1.95 × 10^−3^

**Table 2 sensors-23-09320-t002:** Algorithm complexity.

Algorithm	Algorithm Complexity
FBSS	$𝒪\left(M^{3}+N^{3}\right)$
ESS	$𝒪\left(3M^{3}+3N^{3}\right)$
Proposed	$𝒪\left(2\left(L-\tau \right){\left(M+N\right)}^{2}+{\left(M+N\right)}^{3}+3M^{3}+3N^{3}\right)$

## Data Availability

The data presented in this study are available on request from the corresponding author.

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
