# Peer review of "Coherent DOA Estimation Algorithm with Co-Prime Arrays for Low SNR Signals"

_sensors, 2023, doi:10.3390/s23239320_

Round 1
Reviewer 1 Report
Comments and Suggestions for Authors
In this paper, the authors proposed a DOA estimation algorithm using coprime array under low SNR scenario, where coherent sources are considered. Before this paper is ready to be accepted, I have several minor comments.
(1) The expression "Direction of Arrival (DOA)" should be just used once where it appears for the first time. When it appears again, please just use the abbreviation "DOA" .
(2) To make the equations more comprehensible, please use bold font to represent the vector and matrix, and use the unbold font to represent the scalars.
(3) The authors consider CRB as the benchmark in simulations, which, however, is a local bound. I recommend the authors consider the global tight Ziv-Zakai bound for DOA estimation for a better comparison, where the closed-form solution can be found in the following reference.
[R1] Z. Zhang, Z. Shi and Y. Gu, "Ziv-Zakai Bound for DOAs Estimation," in IEEE Transactions on Signal Processing, vol. 71, pp. 136-149, 2023, doi: 10.1109/TSP.2022.3229946.
Author Response
Thank you for your summary and positive comments on our work. The pdf below is a response to specific comments.

Reviewer 2 Report
Comments and Suggestions for Authors
Thank you for effort and interesting work.
I have some comments that must be considered in the modified manuscript.
-------------------------------------------------------------------------------------
The work in this paper mainly deals low signal to noise ratio systems.
Authors introduced an Enhanced Spatial Smoothing (ESS) algorithm that utilizes space-time correlation matrix for de-noising and decoherence.
NOTES
1) Abstract must have some numerical values, especially when you write (proposed algorithm has been proven to be accurate under low SNR and small snapshots, as well as having high angular resolution). You must write the achieved accuracy and resolution, like that written in "Conclusion" section.
2) Tell us that Eq. (3) is a definition (or: by definition) or, put a reference.
3) Is the approximation mentioned in Eq. (78) a universal? or: use a reference.
4) Eq. (9) needs a reference.
5) Although the mathematical formulation is strong, but, it needs a simple block diagram (or flow chart) that summarizes the steps or procedure. This is clearly found in the table after Eq. (29). Thank you.
6) In both Fig. 3 and Fig. 4, is it better to display results in bars instead of broken lines? i.e. are the values of SNR discrete or continuous?
7) Same remark for Figs. 5+6 (I think the number of snapshots is discrete).
8) Same also for Figs. 7+8+9.
9) Discussion after Table 2 needs percentage values to clarify the superiority of the proposed algorithm as compared to others.
10) The Appendix needs a title.
12) In the References list:
- you must mention all authors names (do not use ....et al.).
- Some references need the source (journal name and details or conference name. Ex: Ref. 1).
- In Refs. 10+19+23, I do not understand (...... n.pag)?
Author Response

(The authors gave the same response as above.)

Reviewer 3 Report
Comments and Suggestions for Authors
Please find the comments referring to the paper as an attachment.

Author Response

(The authors gave the same response as above.)

Round 2
Reviewer 3 Report
Comments and Suggestions for Authors
All comments of the reviewer have been included in the revised version of the paper. I recommend publication of this paper in its current form.